# REM Sleep Stage Identification with Raw Single-Channel EEG

**DOI:** 10.3390/bioengineering10091074

**Published:** 2023-09-11

**Authors:** Gabriel Toban, Khem Poudel, Don Hong

**Affiliations:** 1Computational & Data Science Ph.D. Program, Middle Tennessee State University, Murfreesboro, TN 37132, USA; khem.poudel@mtsu.edu (K.P.); don.hong@mtsu.edu (D.H.); 2Department of Computer Science, Middle Tennessee State University, Murfreesboro, TN 37132, USA; 3Department of Mathematical Sciences, Middle Tennessee State University, Murfreesboro, TN 37132, USA

**Keywords:** DWT, CNN, EEG, sleep stages

## Abstract

This paper focused on creating an interpretable model for automatic rapid eye movement (REM) and non-REM sleep stage scoring for a single-channel electroencephalogram (EEG). Many methods attempt to extract meaningful information to provide to a learning algorithm. This method attempts to let the model extract the meaningful interpretable information by providing a smaller number of time-invariant signal filters for five frequency ranges using five CNN algorithms. A bi-directional GRU algorithm was applied to the output to incorporate time transition information. Training and tests were run on the well-known sleep-EDF-expanded database. The best results produced 97% accuracy, 93% precision, and 89% recall.

## 1. Introduction

Sleep stages are the most-precise way to separate wakefulness from the sleep state. Physiologic and pathologic events can be identified and studied through examining the stages of sleep or sleep stages. Certain health insurances require that wakefulness is separated from the sleep state using an electroencephalogram (EEG) in order for a sleep study to be conducted.

An EEG uses electrodes placed on the head to detect electric potentials in the brain. The difference between the electric potentials of two electrode positions, called an EEG channel or channel, are recorded in regular intervals to produce brainwaves. One electrode is the active electrode, while the other is the reference electrode [1]. The electrode positions are well-defined by the International 10/20 system [1,2,3]. The EEG channels that are best to use were standardized originally by Rechtscheffen and Kales [4], commonly known as the R&K standard. The American Academy of Sleep Medicine (AASM) maintains and updates the R&K standard today.

Sleep stages are typically identified with a combination of parameters known as a polysomnographic record (PSG). These parameters include, but are not limited to, electroencephalogram (EEG) derivations, electrooculogram (EOG) derivations, and a chin electromyogram (EMG) derivations [1]. Elements from all three are used to identify sleep stages [1]. An EEG monitors brainwaves. An EOG monitors eye movements. An EMG monitors muscle movements. EEGs, EOGs, and EMGs all can use electrodes placed on the head and face.

An alternative to a PSG is an at-home study. There can be many electrodes and channels used in a PSG. The AASM recommends that eight electrode positions creating six channels be used for an EEG alone [1]. Using this many channels and the variety of parameters for the PSG require special training to use, special training to interpret, expensive equipment, and, likely, a lab. Some at-home studies use a single-channel EEG, requiring very little training to use, less-expensive equipment, and no lab. The at-home study still requires special training to interpret.

Automated sleep staging can be accomplished with a combination of parameters from a PSG [5,6,7,8,9] or a single-channel EEG [10,11,12,13]. Hong et al. claims support vector machines (SVMs) and artificial neural networks (ANN) are the “tool of choice” for any data that are classified a priori [14]. Ebrahimi et al. achieved 93% accuracy using the wavelet decomposition of a single-channel EEG and a neural network [11]. Poudel et al. was able to compress, denoise, and classify electrocardiograms (ECGs) using discrete wavelet transforms (DWTs) and convolutional neural networks (CNNs) [15]. Kurt et al. achieved 97–98% accuracy using the wavelet decomposition of the EEG, EOG, and chin-EMG [8]. Li et al. enhanced sleep stage N1 classification from 41.5% to 55.65% using preprocessing signals with wavelet threshold denoising (WTD) and wavelet packet transform (WPT) [16]. ElMoaqet et al. achieved an average per-class accuracy of 91.2%, a sensitivity of 77%, a specificity of 94.1%, and a precision of 75.9% preprocessing signals with a wavelet transform and a bidirectional long short-term memory (BiLSTM) [17]. Fu et al. achieved an f1 score of 81.79 preprocessing data using the wavelet threshold method and a bidirectional recurrent neural network [18].

In order to automate sleep staging with an at-home study, the wavelet decomposition of a single-channel EEG is used as a preprocessing step to a CNN to a bidirectional gated recurrent unit (BiGRU) model. Alvarez et al. [13] listed the following algorithms, among others, in order of highest accuracy for sleep staging: ANN, support vector machine (SVM), hidden Markov model, and discriminant analysis. Lotte et al. [19] scored many algorithms for classification in an EEG-based brain–computer interface including linear discriminant analysis, support vector machine (SVM), ANN multilayer perceptron, other ANNs, Bayes quadratic, the hidden Markov model, k-nearest neighbors, the Mahalanobis distance, and combinations. Even though Lotte et al. [19] concluded that the SVM would be the best for the EEG-based brain–computer interface, they also stated that neural networks are the most-used category of classifiers. Seeing that both Alvarez et al. and Lotte et al. put significance on SVMs and hidden Markov models, these two algorithms may be tested in the future.

There are many time–frequency signal analysis representations that could be tested. The Gabor transform, short-time Fourier transform, Wigner distributions, and many others could be tested with future studies. This study used the multilevel discrete wavelet transform.

## 2. Methods

### 2.1. Data

This study used the “Sleep Recording and Hypnograms in European Data Format (EDF)” dataset or “The Sleep-EDF Database [Expanded]” from physionet.org [20]. The portion of the database used was from a study on healthy patients from 1987–1991. There were 20 patients available. There were 10 males and 10 females ranging from 25–34 years old.

Each patient had two relevant files: a polysomnogram (PSG) recording in EDF format and a hypnogram of annotations in EDF+. EDF is a standard format for exchanging EEG recordings [21]. EDF+ has all the capabilities of EDF and the ability to contain annotations [21]. The PSG recording included an EEG from the Fpz-Cz and Pz-Oz electrode locations, an EOG (horizontal), a submental chine EMG, an event marker, an oral–nasal respiration, and rectal body temperature. The hypnogram is an annotation of sleep patterns. The annotations included W for wakefulness, R for REM, 1 for Stage 1, 2 for Stage 2, 3 for Stage 3, 4 for Stage 4, M for movement time and ? for not scored. Once the first annotation was reached, the rest of the annotations were 30 s apart, creating the 30 s epochs.

The international 10/20 system uses labels with letters and numbers to identify the different electrode positions on the head, as referenced in Figure 1. The letters identify the brain lobe that the electrode is over, as identified in Table 1. The numbers identify the position in the direction going from one ear to the other. To find each position, a technician begins at a starting position and moves around the head in percentages [2,3].

Different electrode positions and channels detect different brainwaves, activities, and events. Table 2 lists some of the brainwaves, activities, and events and which lobe position or channel detects them. V-waves, K-complexes, spindles, sawtooth waves, and Alpha waves are all at least adequate in the frontal and central lobes of the brain. This is why we chose the Fpz-Cz channel.

Special software is required to read the EDF and EDF+ files. EDFbrowser is a free EDF and EDF+ reader, which can export the contents to a text file. All the contents of the PSG and hypnogram file were exported to a text file with the option to have time in seconds from the first recording.

The PSG and hypnogram text files were imported into a custom python 3 script. The python 3 script, epochs.py, uses the pywavelets package to complete a wavelet decomposition of any suggested level. The output file contains the annotation and decomposition of each epoch per line unless the annotation is for movement time or not scored. Movement time and not scored epochs are skipped. All the awake epochs except the last one are removed prior to the first epoch that is not a wake epoch. The count of each annotation per file is printed to standard out.

A focus of this study was meaningful interpretable information. We focused on 1 sleep stage at a time to minimize confusion in interpretability. We focused on REM sleep because of its relationship with degenerative brain disease [22,23]. All epochs were labeled either REM or non-REM.

### 2.2. Wavelet Decomposition

The EEG had a recording every 10 ms. There were 3000 recordings per epoch. This implies that the maximum decomposition level is 8. Each decomposition level produced a different number of coefficients, as listed in Table 3.

During the decomposition process, the wavelet functions were convolved with the signal at different scales and positions, resulting in a set of coefficients that represent the contribution of each wavelet function to the signal at each scale and position. The coefficients at each scale represent a different frequency band of the signal. After each convolution, the resulting coefficients are downsampled, which means that the number of samples is reduced by a factor of two by discarding redundant samples. The sampling rate after each level of decomposition can be calculated using the following formula, where the original signal has a sample rate of $f_{s}$ and *n* is the level of decomposition.
(1)$$f_{sn}=\frac{f_{s}}{2^{n}}$$

The resulting sample rates are matched to the EEG frequencies in Table 4.

The reconstructed signal after each of the five DWT levels on a 30 s epoch sample is visualized in Figure 2.

### 2.3. Model

Each of these reconstructed signals was used as the input to a CNN. The hyperparameters of the neural network were tested using a custom hyperparameter random search. The random search was divided into 4 steps: preprocessing/training parameters, the candidate builder, the model builder, and the model trainer/tester. These were managed with the random search python script.

#### 2.3.1. Preprocessing/Training Parameters

The random search starts building a model by defining a few parameters based on the preprocessing of the data and the training of the model. The preprocessing parameters included the number and set of input files, the normalization standard deviation and mean, the mother wavelet, the decomposition level, and the expected frequency band. The training parameters included the number of cross-validation folds, the validation split percentage, the max number of training epochs, and the batch size. The following is a sample of the preprocessing and training parameters.


**dataFiles:**
input152.csv,input042.csv
**cvFolds:**
10
**validation_split:**
0.1
**epoch:**
100
**batchSize:**
381
**normSTD:**
19.476044983423417
**normMean:**
0.017218097667286797
**mother wavelet:**
db2
**decomp level:**
5
**frequency band:**
Delta

The data files were preprocessed into 30 s epochs with labels. All the movement time was removed. All wake stages except the last one before the first stage that was not wake were removed. The files use the following naming format with the patient number in 2 digits (PP) and the night number in 1 digit (n).
(2)inputPPn.csv

The other preprocessing arguments were the normalization standard deviation (normSTD), the normalization mean (normMean), the mother wavelet used in the multilevel discrete wavelet transform (mother wavelet), the decomposition level used in the multilevel discrete wavelet transform (decomp level), and the frequency band being represented by this CNN. The normalization process was calculated across the entire dataset in the dataFiles parameter. The mother wavelet was manually chosen.

The training arguments included the batch size of each epoch (batchSize), the maximum number of epochs for training (epoch), the number of cross-validation folds (cvFolds), and the validation split percentage in decimals (validation_split). The batch size was calculated to be the entire training dataset given the record count in the dataFiles (R), the number of cross-validation folds (C), and the validation split (V) with the following formula. The best epoch size was found through testing. The validation split and cross-validation folds were chosen as a baseline for to show the generalization.
(3)batchSize=int((R∗(1−V))/C+1)

#### 2.3.2. The Candidate Builder

The candidate builder creates a json structure Listing 1 with all the necessary information to build a model. Each CNN candidate is expected to have a CNN layer followed by a collection of dense layers. Hyperparameters specific to the CNN include the number of filters (defaulted to 5 or 10) and the kernel size (defaulted to 3, 5, 10, 20, or 66). The number of dense layers and the number of nodes in each layer is a hyperparameter. The activation function, kernel initializers, bias initializers, and optimizers are also all hyperparameters. The following is an example candidate in JSON structure.

**Listing 1.** A sample JSON structure created by the candidate builder and read by the model builder.

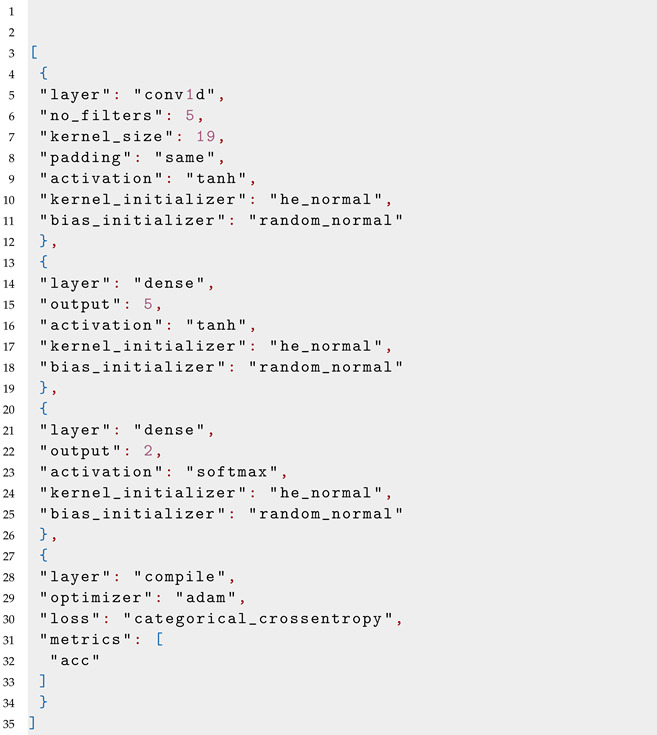



#### 2.3.3. The Model Builder

The model builder reads the json structure Listing 1 and creates a Tensorflow Keras model. A sample CNN model is shown in Figure 3.

#### 2.3.4. The Model Trainer/Tester

The model trainer/tester trains the model, tests the model, and saves all relevant data. The stopping condition is always based on the validation loss. There is an option to explore the models with a tensor board. Each cross-validation training and test set was created using a the Scikit-Learn StratifiedKFold function. Each training set was trained using the batch size, validation percentage, and maximum number of epochs parameters. This training was allowed to save a large collection of metrics including all the fit history, a plot of the loss versus validation loss, and a plot of the area under the ROC curve versus the validation area under the ROC curve. The fit history can include the accuracy, precision, recall, f1 score, and area under the ROC curve. Each test set was then used to obtain a collection of metrics, which was always saved. The test set collection of metrics included true positives, false negatives, false positives, true negatives, true positives, the accuracy, the sensitivity, the specificity, the recall, the precision, the f1 score, and the last loss calculated in the fit history.

### 2.4. Filter Comparisons: The Frequency Band Wavelet Model

Once all the models were trained, the CNN filters were extracted and examined for interpretability. Each frequency band representative model trained on the data decomposed by each mother wavelet (the frequency band wavelet model) produced a different set of filters in each cross-validation step. The filters extracted were the filters from the first cross-validation step with the highest f1 score for each frequency band wavelet model. Two methods of comparison were used: visual comparison and Spearman rank correlation.

The visual comparison was intended to see if there was any relationship with known clinical markers provided by the AASM. The clinical markers included, but were not limited to k-complexes, sleep spindles, and slow wave activity. K-complexes are sharp negative EEG waves that occur during N2 sleep in response to external stimuli. The presence of K-complexes can be used as a marker of sleep depth and stability. Sleep spindles are brief bursts of EEG activity in the frequency range of 11–16 Hz that occur during N2 sleep. The presence and density of sleep spindles can be used as a marker of sleep quality. Slow wave activity (SWA) is EEG activity in the frequency range of 0.5–4 Hz that occurs during N3 sleep. The amount of SWA is an important marker of sleep quality, with higher levels of SWA indicating deeper and more-restorative sleep.

The Spearman rank correlation is used as a first step numerical analysis of filters. The assumption is that a filter can be found by every wavelet model. Spearman’s rank correlation is calculated for the filters from each extracted frequency band wavelet model against each of the other extracted filters for each wavelet model associated with the same frequency band. For example, the Theta coiflet 4 model filters were compared against all the other models representing the Theta frequency band. There are 5 or 10 filters for each model. An example single comparison would be the 1st filter of the Theta coiflet 4 model against the 10th filter of the Theta Haar model.

Spearman’s rank coefficient was then used to find filters that were found by other training. A formula was used to find the filters with the most and highest correlations. The formula uses the 95th percentile of all the Spearman’s rank coefficients as a threshold, the filter threshold. The top 5 filters with the most Spearman’s rank coefficients greater than the filter threshold were found to be the most-relevant filters.

## 3. Results

There are tens of gigabytes of results that are available for review in a result set. Things such as the exact model definition, the training parameters, and the results of each training can be found in the result set. The following is a collection of sample and high-priority results. The wavelet model with the highest f1 score is coiflet 4. The examples are primarily based on that. The results described below include: the training parameters, the fit history with the validation of a single training, the cross-validation results of a single training, the average cross-validation results for all wavelet models, a plot of filters from the coiflet 4 model and none model, the plot of of the top five correlated filters from each frequency band, and a description of those comparisons.

### 3.1. The Model

The model used is a variation on the most-common models, the CNN to the RNN models [24,25,26,27,28]. The model in Figure 4 consists of a convolution part and a recurrent or a memory part. The input data are three epochs: a pretarget epoch, a target epoch, and a post-target epoch. Each of the three epochs is fed through wavelet decomposition based on the five frequency bands. The decomposed signal based on each frequency band from each epoch is input for the CNN model. The output from all frequency bands and epochs is concatenated. There are outputs from 15 datasets fed through the CNN model (five frequency bands for each of the three models). The concatenated output is input to the BiGRU model. The resulting model is represented in Figure 4.

The convolutional part consisted of five CNNs. The GACNN model [29] found that the most information can be extracted from an EEG signal using four different filter sizes for each CNN. This model adds an additional CNN with a different filter size. There is 1 CNN for each of the Delta, Theta, and Alpha EEG bands and 2 CNNs for the beta EEG band for a total of 5 CNNs. Each CNN has a filter size that allows the frequency of the filter to fit within the specified frequency band, as described in Table 5. The beta EEG band spans the largest frequency and is at the boundary of the sampling rate, which can lead to noise in the signal. Splitting the beta EEG into 2 CNN filter sizes is an attempt to separate out the noise and extract more-interpretable information.

Each CNN was trained independently on the 30 s epochs to score REM and not REM. Each CNN was associated with a different frequency range in an EEG described in Table 5. Each CNN had either 5 or 10 filters. The pooling layers pool size is on the left and the factor by which to downscale for strides is on the right in Figure 4. In each CNN, the same activation, weight initializer, and bias initializer are used on each layer except the final layer, which always uses softmax. The activation function is always tanh. The weight initializer and bias initializer is described in Table 5.

The output of the five CNNs are concatenated without any adjustment. A new CNN dataset is created with each record consisting of three epochs of the concatenated output of the CNNs. Each record is labeled with the middle or current epoch. This duplicates the pre- and post-epochs used to score the current epoch by technicians.

The RNN has 2 layers of bidirectional gated recurrent unit (GRU) cells, 3 fully connected layers, and the final softmax layer. The input to the first GRU is the new CNN dataset. As can be seen in Figure 4, the first GRU layer outputs 104 dimensions and the second 302 dimensions. All the other parameters were set to the default Keras parameters. No optimization was required.

#### 3.1.1. Training Parameters

The first part of the model completed through the random search is the model for the Delta frequency band. The preprocessing and training arguments are as follows:
**dataFiles:**input152.csv,input042.csv,input171.csv,input161.csv,input091.csv,input002.csv,input142.csv,input031.csv,input082.csv,input151.csv,input101.csv,input032.csv**cvFolds:**10**validation_split:**0.1**epoch:**100**batchSize:**2054**normSTD:**20.366845241085922**normMean:**−0.4573919127663568**mother wavelet:**coif4**decomp level:**5**frequency band:**Delta


#### 3.1.2. Fit History Data

The training then creates the following “fit history” Table 6 and Table 7.

The fit history for the training data showed that the model prediction was 100% on all metrics while the loss started low and steadily decreased. The fit history for the validation started with all metrics in the 96% range and moving up, while the loss started low and steadily decreased. This suggested that the model was accurate with the ability to generalize.

#### 3.1.3. Fit History Plots

This fit history was then plotted in two different plots: loss vs. validation loss and auc_roc vs. validation auc_roc. See Figure 5.

This visualization of the fit history is a simpler way to confirm that this model was a good fit for these data.

#### 3.1.4. Cross-Validation

After the training, the cross-validation test data were used to create a different set of metrics. These cross-validation data are aggregated across all cross-validation folds in Table 8.

Table 8 identifies the true positives, false positives, false negatives, and true negatives. Through 6 of the 10 cross-validation fold, there were 0 false positives and false negatives. Table 8 identifies that these six folds were 100% accurate and the average accuracy was 97%, despite that the f1 score became as low as 11% and the average f1 score was 85%. This was not as good as the other models, and the other parts of this complete model need to be added to improve the model.

#### 3.1.5. Average Cross-Validation

The same model was tested with wavelet decomposition using 28 different mother wavelets and 1 without any wavelet decomposition. All the results produced 98% accuracy or higher accuracy with an f1 score between 93% and 94%. Table 9 and Table 10 show the average cross-validation results from all 22 mother wavelets. The coif4 mother wavelet had the highest f1 score of 0.9456. The db5 mother wavelet had the lowest f1 score of 0.9374. The model without any wavelet decomposition was closer to the lowest f1 score of 0.9392, but notably not the lowest.

Table 11 shows the comparison to the other models. The models used for comparison used the same dataset. These models attempted to score all stages of sleep, and while this could be the end goal, focusing on one stage of sleep can help the model be more explainable. In addition, other stages of sleep can have very specific clinical markers, which can be easily identified with simpler methods.

#### 3.1.6. CNN Filters

The filters from each frequency band wavelet model were extracted and plotted for visual inspection. The clinical markers such as k-complexes, v-waves, and spindles can be seen in the filters. Slow waves cannot be seen as easily. Figure 6 has the filters from the coiflet 4 model and the none model (the model trained on data without preprocessing). The CNN filter size for each band was Delta (35), Theta (19), Alpha (10), Beta1 (5), and Beta2 (2). The CNN filter size, fs, was calculated based on a representative frequency within each frequency band rf and the sample rate, sr, for this dataset before preprocessing (100 Hz).
(4)$$fs=\frac{sr}{rf}=\frac{100}{rf}$$

Spearman’s rank coefficient was then calculated for all wavelet models in the same frequency band. This coefficient has more implications for the Delta, Theta, and Alpha frequency bands because of their filter size. The 95th percentile for the Spearman’s rank coefficient was 0.34145658263305323. The percent of coefficients for the top five of each frequency band that were higher than the 95th percentile are in Table 12.

Each of the top five models were then plotted in Figure 7 to be visually inspected. The clinical markers such as k-complexes, v-waves, and spindles can be seen in the filters. Slow waves cannot be seen as easily.

## 4. Discussion

Preprocessing with wavelet decomposition provided marginal improvement in the f1 score in most cases. From the visual inspection, clinical markers can be seen in the CNN filters. Preprocessing with wavelet decomposition does not definitively improve the ability to see those clinical markers. Similarity comparisons between the filters found that bigger filters tended to be less similar. Table 13 compares the top 5 f1 scores with the top 1 Pearson’s coefficient in each frequency band.

In future studies, every aspect of this study could be changed to improve the necessary results: the choice of EEG channel; the choice of sleep channel; the choice of sleep stage; the choice of epoch size. Of the time–frequency techniques, the wavelet transform is the best choice, but the modification of the wavelet transform should be changed to possibly a continuous wavelet transform or a new modification of the discrete wavelet transform. The support vector machine, multilayer perceptron, convolution neural networks, deep feed-forward neural networks, and the hidden Markov model should all be tested for classification.

This study was limited to the choice between two sleep channels. For at-home studies, there are different devices that use different channels. One device uses two electrodes around the Fpz electrode position. This position is basically on the forehead of a person and would be one of the easiest positions to locate for an untrained person.

The goal of this paper was to support an at-home sleep study. A sleep study needs to be able to detect all stages, but it is very important for a sleep study to identify sleep from wakefulness. If REM cannot be identified, all stages cannot be identified with a single-channel EEG.

The epoch was originally defined based on the ease of use. The complications of the paper size and time required for an analysis can be eliminated by the choice of epoch size. A training dataset may be difficult to find for a small epoch size, but each individual wave can be classified based on the definitions provided by the AASM. This could result in epochs being scored literally by which waves make up the majority of the epoch.

Time–frequency analysis has many new modifications for discrete wavelet transforms. These modifications should be tested for improvements to the current discrete wavelet transforms. Furthermore, the continuous wavelet transform (CWT) can contain more detail than the discrete wavelet transform (DWT). The CWT should be tested against the new modifications of the DWT.

## Figures and Tables

**Figure 1 bioengineering-10-01074-f001:**
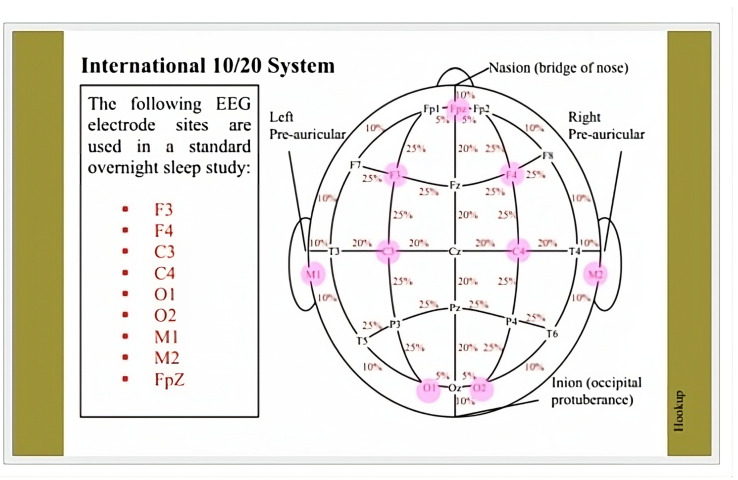
A representation of the electrode positions identified by the 10/20 system. Each position is labeled by 1 to 3 letters and/or a number, which are listed on the left side of this image. Image courtesy of https://sleeptechstudy.wordpress.com/category/1020-system/ [3], accessed on 4 July 2017.

**Figure 2 bioengineering-10-01074-f002:**
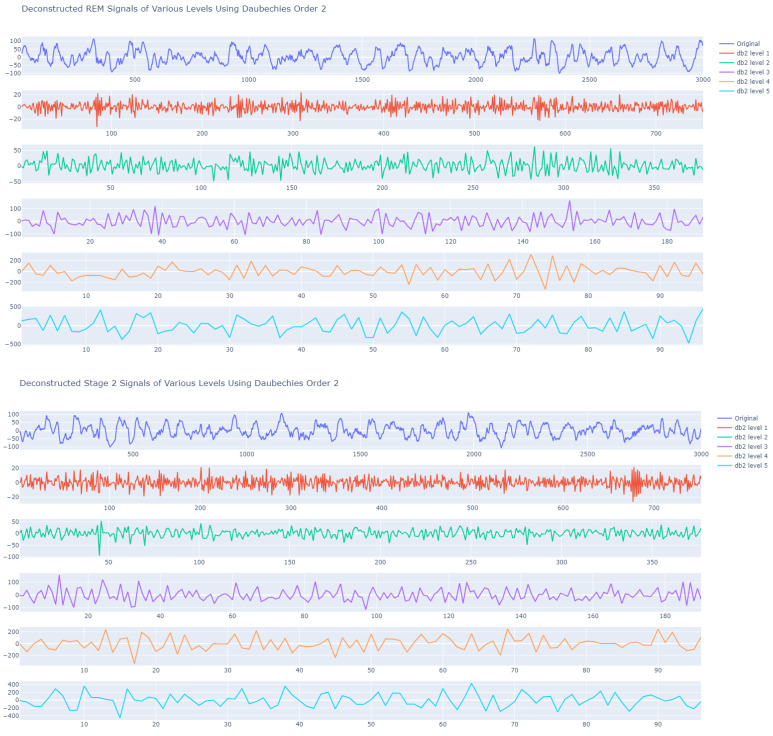
Each series of plots shows a different stage decomposed into 5 levels of wavelet decomposition using the Daubechies order-2 mother wavelet. Each plot in the series shows a 30 s epoch sample at a different of level of wavelet decomposition using the Daubechies order-2 mother wavelet. The top plot is the original signal. Each one down after that goes to a higher level of decomposition starting at Level 1 and going to Level 5, respectively.

**Figure 3 bioengineering-10-01074-f003:**
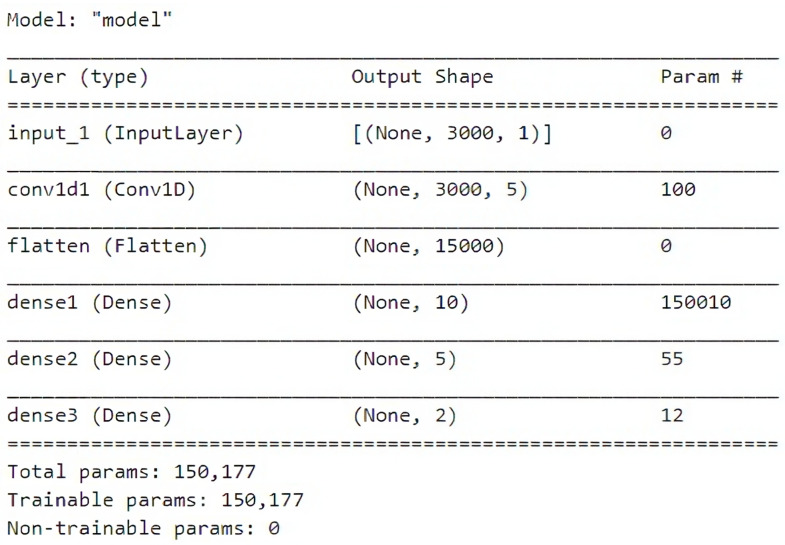
The output of model.summary() from Tensorflow Keras of a model built by reading a candidate builder json structure.

**Figure 4 bioengineering-10-01074-f004:**
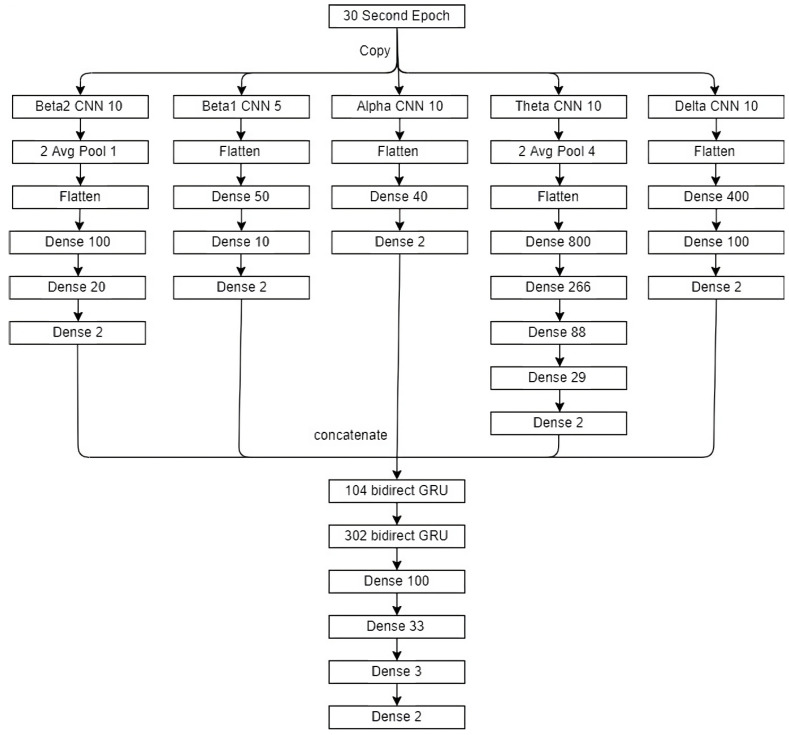
The figure labels the model input, the 5 convolutional networks applied to each input, and the recurrent network applied to the 3 time steps of the input. The input is a 30 s epoch. The 30 s epoch is copied to provide input to each of the 5 convolutional networks. The first box of each convolutional network names the EEG band it represents and labels the number of filters. Each box following represents different information about the layer in the network it represents. Boxes about pooling layers define the scaling factor, pooling method, and stride. Boxes about dense layers and recurrent layers define the output size. Concatenate is meant to identify the concatenation of the output of all convolutional networks, as well as the concatenation of the 3 input time steps.

**Figure 5 bioengineering-10-01074-f005:**
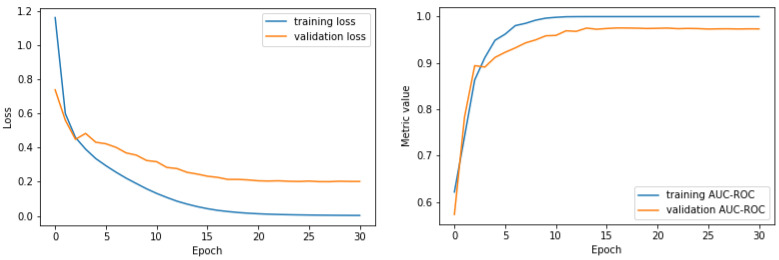
The loss and validation loss of the first cross-validation fold with data that were preprocessed with the multilevel discrete wavelet transform Level 5 with the mother wavelet Daubechies order 2. The AUC ROC and validation AUC ROC of the first cross-validation fold with data that were preprocessed with the multilevel discrete wavelet transform Level 5 with the mother wavelet Daubechies order 2.

**Figure 6 bioengineering-10-01074-f006:**
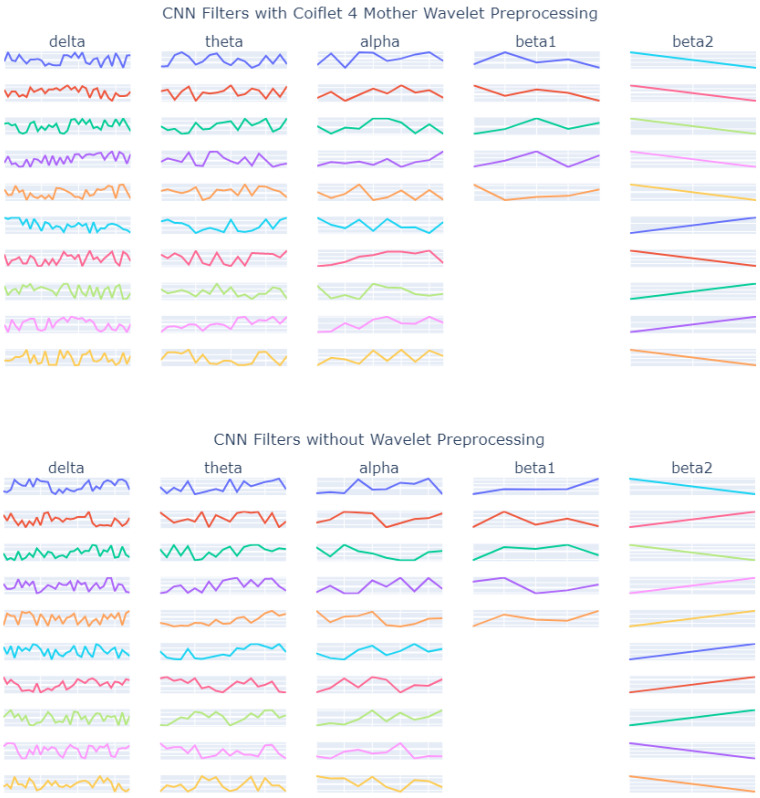
Filters from the coiflet 4 wavelet model and the no wavelet model are plotted. The filters were from the weights created during the cross-validation with the highest f1 score. Clinical markers such as k-complexes, v-waves, and spindles can be seen in the filters. Slow waves cannot be seen as easily.

**Figure 7 bioengineering-10-01074-f007:**
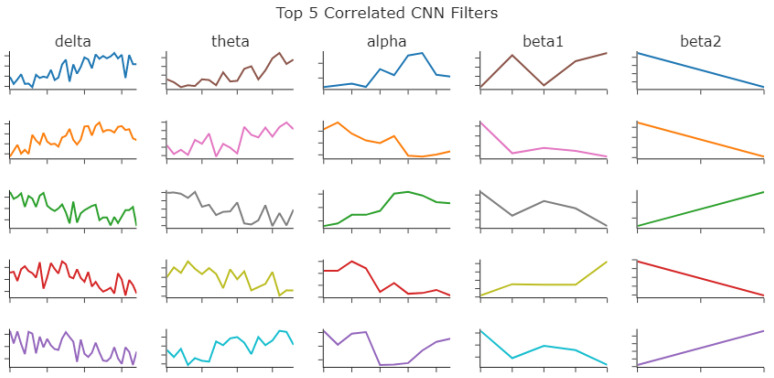
The plot of the top 5 wavelet models as identified with the calculation represented in Table 12.

**Table 1 bioengineering-10-01074-t001:** The brain region that relates to every electrode location [3].

Abbreviation	Brain Lobe/Region
Fp	Frontal Pole
F	Frontal
T	Temporal
C	Central
P	Parietal
O	Occipital
M	Mastoid

**Table 2 bioengineering-10-01074-t002:** The brain regions/electrode channels where brainwaves, activities, and events (markers) are most likely to appear. To understand what stages are identified with each marker, reference the AASM scoring guide [1]. A blank space means there were no specifications for this region/channel and marker combination. “Yes” means that this channel does identify slow waves, while “No” means that the channel does not identify slow waves. “Maximal” implies the maximal wave amplitude will be recognized. “Adequate” means adequate for identification. All information except “Adequate” was gathered from the AASM scoring guide [1]. “Adequate” comes from the R&K standard [4]. E1-Fpz refers to the frontal regions referenced to the contralateral ear or mastoid.

	Slow Wave	V-Waves	K-Complexes	Spindles	Sawtooth	Alpha
Frontal			Maximal			
Central		Maximal		Maximal	Maximal	Adequate
Occipital						Maximal
Fz-Cz	No					
E1-Fpz	Yes					

**Table 3 bioengineering-10-01074-t003:** The number of coefficients produced for a 30 s epoch at each available level of decomposition.

Level	Number of Coefficients
8	14
7	26
6	49
5	96
4	190
3	377
2	752
1	1501

**Table 4 bioengineering-10-01074-t004:** DWT frequency bands and sample rates.

Frequency Band	DWT Level	Sample Rate (Hz)
Beta2 (>13 Hz)	1	50.0
Beta1 (>13 Hz)	2	25.0
Alpha (8–13 Hz)	3	12.5
Theta (4–7.99 Hz)	4	6.25
Delta (0–3.99 Hz)	5	3.125

**Table 5 bioengineering-10-01074-t005:** Each convolutional network represents a different frequency band. The Delta, Theta, and Alpha frequency bands are gathered from common definitions. The beta frequency band is divided into 2 bands because it is the largest frequency band, and noise in a signal can be found in the highest frequencies. A midpoint value in the frequency band is chosen to be the CNN filter frequency. The CNN filter size is calculated using the CNN filter frequency and the sample rate of the data set. CNN Filter Size ≈ (sample rate)/(CNN Filter).

EEG Frequency Band (Hz)	CNN Filter Frequency (Hz)	CNN Filter Size (Samples)
Delta (0–3.99)	2.8	35
Theta (4–7.99)	5.2	19
Alpha (8–13)	10	10
Beta1 (>13)	20	5
Beta2 (>13)	50	2

**Table 6 bioengineering-10-01074-t006:** Fit history.

Loss	Accuracy	Precision	Recall	f1_SCORE	auc_roc
0.0018	1	1	1	1	1
0.0012	1	1	1	1	1
0.0008	1	1	1	1	1
0.0007	1	1	1	1	1
0.0006	1	1	1	1	1
0.0005	1	1	1	1	1
0.0005	1	1	1	1	1

**Table 7 bioengineering-10-01074-t007:** Validation fit history.

val_loss	val_accuracy	val_precision	val_recall	val_f1_score	val_auc_roc
0.1018	0.9606	0.9606	0.9606	0.9606	0.9945
0.0978	0.9685	0.9685	0.9685	0.9685	0.9947
0.0797	0.9790	0.9790	0.9790	0.9790	0.9967
0.0773	0.9790	0.9790	0.9790	0.9790	0.9967
0.0812	0.9764	0.9764	0.9764	0.9764	0.9963
0.0844	0.9738	0.9738	0.9738	0.9738	0.9960
0.0843	0.9738	0.9738	0.9738	0.9738	0.9960

**Table 8 bioengineering-10-01074-t008:** Cross-validation results and the average across those cross-validations with data that were preprocessed with the multilevel discrete wavelet transform Level 5 with the mother wavelet Daubechies order 2. The results identify the true positives, false positives, true negatives, false negatives, accuracy, specificity, recall, precision, and f1 score of each cross-validation test set.

CV Fold	True REM	False REM	False NonREM	True NonREM	
1	7	17	275	1983	
2	228	11	54	1989	
3	227	0	55	2000	
4	223	63	59	1937	
5	214	0	67	2000	
6	270	2	12	1997	
7	279	8	3	1991	
8	278	0	4	1999	
9	281	1	1	1998	
10	99	25	183	1974	
Average	210.6	12.7	71.3	1986.8	
CV Fold	Acc	Spec	Recall	Precision	f1 Score
1	0.872	0.992	0.025	0.292	0.046
2	0.972	0.995	0.809	0.954	0.875
3	0.976	1	0.805	1	0.892
4	0.947	0.969	0.791	0.78	0.785
5	0.971	1	0.762	1	0.865
6	0.994	0.999	0.957	0.993	0.975
7	0.995	0.996	0.989	0.972	0.981
8	0.998	1	0.986	1	0.993
9	0.999	0.999	0.996	0.996	0.996
10	0.909	0.987	0.351	0.798	0.488
Average	0.9633	0.9937	0.7471	0.8785	0.7896

**Table 9 bioengineering-10-01074-t009:** The average cross-validation results with the confidence interval of the model trained with 29 different datasets. Twenty-eight of the datasets were preprocessed by twenty-eight different mother wavelets. One did not have any preprocessing. These are listed from highest f1 score to lowest f1 score, with the one without preprocessing highlighted in red.

Wavelet	Acc	Spec	f1 Score
coif4	0.991 (0.9711, 1.0109)	0.9996 (0.9987, 1.0005)	0.9456 (0.8253, 1.0659)
sym3	0.9911 (0.9715, 1.0107)	0.9999 (0.9997, 1.0001)	0.9451 (0.8237, 1.0665)
db4	0.9911 (0.9715, 1.0107)	0.9999 (0.9997, 1.0001)	0.9448 (0.8232, 1.0664)
Haar	0.9911 (0.9715, 1.0107)	1 (1, 1)	0.9445 (0.8215, 1.0675)
coif5	0.991 (0.9711, 1.0109)	1 (1, 1)	0.9441 (0.8202, 1.068)
db6	0.99 (0.9708, 1.0092)	0.9981 (0.9959, 1.0003)	0.941 (0.8241, 1.0579)
sym7	0.9902 (0.9705, 1.0099)	0.9993 (0.9979, 1.0007)	0.941 (0.8177, 1.0643)
sym4	0.9897 (0.9708, 1.0086)	0.9981 (0.9957, 1.0005)	0.9407 (0.8248, 1.0566)
sym8	0.99 (0.9708, 1.0092)	0.9985 (0.9969, 1.0001)	0.9407 (0.8212, 1.0602)
db7	0.9901 (0.9704, 1.0098)	0.999 (0.9974, 1.0006)	0.9401 (0.817, 1.0632)
db1	0.9896 (0.97, 1.0092)	0.998 (0.9958, 1.0002)	0.9397 (0.8201, 1.0593)
sym6	0.9898 (0.9704, 1.0092)	0.9984 (0.9965, 1.0003)	0.9397 (0.8191, 1.0603)
db2	0.9899 (0.9705, 1.0093)	0.9985 (0.9967, 1.0003)	0.9397 (0.8187, 1.0607)
sym2	0.9899 (0.9704, 1.0094)	0.9986 (0.9964, 1.0008)	0.9397 (0.8176, 1.0618)
coif2	0.9901 (0.9704, 1.0098)	0.9988 (0.9972, 1.0004)	0.9397 (0.8167, 1.0627)
sym5	0.9898 (0.9699, 1.0097)	0.9984 (0.9968, 1)	0.9394 (0.8165, 1.0623)
None	0.9894 (0.9691, 1.0097)	0.9977 (0.9954, 1)	0.9392 (0.8173, 1.0611)
coif1	0.9898 (0.9699, 1.0097)	0.9985 (0.9969, 1.0001)	0.9391 (0.8155, 1.0627)
gaus1	0.9892 (0.9691, 1.0093)	0.9976 (0.995, 1.0002)	0.939 (0.8188, 1.0592)
db8	0.9896 (0.9698, 1.0094)	0.9983 (0.9967, 0.9999)	0.9388 (0.8155, 1.0621)
db3	0.9895 (0.9696, 1.0094)	0.9983 (0.9963, 1.0003)	0.9387 (0.8163, 1.0611)
gaus3	0.9892 (0.9697, 1.0087)	0.9976 (0.9952, 1)	0.938 (0.8193, 1.0567)
mexh	0.989 (0.9685, 1.0095)	0.9973 (0.9945, 1.0001)	0.938 (0.8165, 1.0595)
dmey	0.9895 (0.9696, 1.0094)	0.9981 (0.9959, 1.0003)	0.938 (0.815, 1.061)
gaus2	0.9891 (0.9683, 1.0099)	0.9976 (0.9948, 1.0004)	0.938 (0.8148, 1.0612)
gaus4	0.9891 (0.9686, 1.0096)	0.9975 (0.995, 1)	0.9378 (0.8149, 1.0607)
coif3	0.9891 (0.9696, 1.0086)	0.998 (0.9956, 1.0004)	0.9377 (0.8175, 1.0579)
morl	0.9889 (0.9687, 1.0091)	0.9976 (0.995, 1.0002)	0.9376 (0.8165, 1.0587)
db5	0.9892 (0.9696, 1.0088)	0.9979 (0.9955, 1.0003)	0.9374 (0.8168, 1.058)

**Table 10 bioengineering-10-01074-t010:** The average cross-validation results with the confidence interval of the model trained with 29 different datasets. Twenty-eight of the datasets were preprocessed by twenty-eight different mother wavelets. One did not have any preprocessing. These are listed from highest f1 score to lowest f1 score, with the one without preprocessing highlighted in red.

Wavelet	Recall	Precision	f1 Score
coif4	0.9294 (0.7742, 1.0846)	0.9922 (0.9755, 1.0089)	0.9456 (0.8253, 1.0659)
sym3	0.9283 (0.7706, 1.086)	0.9984 (0.9956, 1.0012)	0.9451 (0.8237, 1.0665)
db4	0.9279 (0.7703, 1.0855)	0.9973 (0.9921, 1.0025)	0.9448 (0.8232, 1.0664)
haar	0.9276 (0.7683, 1.0869)	1 (1, 1)	0.9445 (0.8215, 1.0675)
coif5	0.9273 (0.7674, 1.0872)	1 (1, 1)	0.9441 (0.8202, 1.068)
db6	0.9297 (0.7762, 1.0832)	0.9837 (0.966, 1.0014)	0.941 (0.8241, 1.0579)
sym7	0.9266 (0.7668, 1.0864)	0.9945 (0.9847, 1.0043)	0.941 (0.8177, 1.0643)
sym4	0.9298 (0.7763, 1.0833)	0.9855 (0.9697, 1.0013)	0.9407 (0.8248, 1.0566)
sym8	0.9284 (0.7717, 1.0851)	0.9895 (0.9788, 1.0002)	0.9407 (0.8212, 1.0602)
db7	0.9258 (0.7662, 1.0854)	0.9934 (0.9826, 1.0042)	0.9401 (0.817, 1.0632)
db1	0.9291 (0.7732, 1.085)	0.983 (0.9651, 1.0009)	0.9397 (0.8201, 1.0593)
sym6	0.9279 (0.7703, 1.0855)	0.9874 (0.9733, 1.0015)	0.9397 (0.8191, 1.0603)
db2	0.9276 (0.7693, 1.0859)	0.99 (0.9779, 1.0021)	0.9397 (0.8187, 1.0607)
sym2	0.9273 (0.7681, 1.0865)	0.9908 (0.9758, 1.0058)	0.9397 (0.8176, 1.0618)
coif2	0.9261 (0.7664, 1.0858)	0.9923 (0.9815, 1.0031)	0.9397 (0.8167, 1.0627)
sym5	0.9276 (0.7683, 1.0869)	0.9871 (0.9751, 0.9991)	0.9394 (0.8165, 1.0623)
None	0.9291 (0.7732, 1.085)	0.9764 (0.9466, 1.0062)	0.9392 (0.8173, 1.0611)
coif1	0.9273 (0.7674, 1.0872)	0.9882 (0.9771, 0.9993)	0.9391 (0.8155, 1.0627)
gaus1	0.9294 (0.7742, 1.0846)	0.9767 (0.9495, 1.0039)	0.939 (0.8188, 1.0592)
db8	0.9273 (0.7681, 1.0865)	0.9846 (0.9696, 0.9996)	0.9388 (0.8155, 1.0621)
db3	0.9268 (0.7687, 1.0849)	0.9843 (0.9671, 1.0015)	0.9387 (0.8163, 1.0611)
gaus3	0.9283 (0.7734, 1.0832)	0.9786 (0.9576, 0.9996)	0.938 (0.8193, 1.0567)
mexh	0.9294 (0.7742, 1.0846)	0.9718 (0.9361, 1.0075)	0.938 (0.8165, 1.0595)
dmey	0.9269 (0.7678, 1.086)	0.9841 (0.967, 1.0012)	0.938 (0.815, 1.061)
gaus2	0.928 (0.7714, 1.0846)	0.9743 (0.9391, 1.0095)	0.938 (0.8148, 1.0612)
gaus4	0.9287 (0.7719, 1.0855)	0.9738 (0.9413, 1.0063)	0.9378 (0.8149, 1.0607)
coif3	0.9263 (0.7701, 1.0825)	0.9827 (0.9637, 1.0017)	0.9377 (0.8175, 1.0579)
morl	0.9274 (0.7719, 1.0829)	0.9757 (0.9463, 1.0051)	0.9376 (0.8165, 1.0587)
db5	0.9265 (0.7692, 1.0838)	0.9837 (0.9666, 1.0008)	0.9374 (0.8168, 1.058)

**Table 11 bioengineering-10-01074-t011:** Each model represented here posted calculated results using the same type of output and the same dataset. While some models did look for all sleep stages and used additional datasets, this table only shows comparisons of REM vs. not REM on the Sleep-EDF Database Expanded [27,29,30].

Model	Accuracy	Precision	Recall	f1 Score
GACNN	91.00	92.00	91.00	91.00
DeepSleepNet		80.90	83.90	82.40
SleepEEGNet		81.63	88.71	85.02
This Model	98.94	99.77	98.94	93.92

**Table 12 bioengineering-10-01074-t012:** The frequency band wavelet models that have the biggest percent of Spearman’s coefficients above the 95th percentile of Spearman’s coefficients. The 95th percentile of Spearman’s coefficients was calculated across all models. Every wavelet model in each frequency band was then ranked by the number of wavelet models for which it had a higher Spearman’s coefficient compared to the 95th percentile.

Freq Band	Wavelet	95th%	% above 95th
Delta	1st db8	0.3415	15.17
Delta	9th sym8	0.3415	14.83
Delta	3rd db3	0.3415	11.72
Delta	6th sym5	0.3415	11.72
Delta	4th db7	0.3415	11.38
Theta	4th None	0.4684	11.72
Theta	9th db8	0.4684	11.03
Theta	7th sym6	0.4684	11.03
Theta	4th sym2	0.4684	10.69
Theta	6th mexh	0.4684	10.00
Alpha	6th db4	0.6485	10.00
Alpha	5th coif2	0.6485	10.00
Alpha	0th sym5	0.6485	10.00
Alpha	0th dmey	0.6485	10.00
Alpha	5th gaus3	0.6485	9.31
beta1	0th db2	0.9000	3.45
beta1	1st sym7	0.9000	3.45
beta1	1st coif4	0.9000	3.45
beta1	0th None	0.9000	3.45
beta1	3rd db7	0.9000	3.45
beta2	9th None	1.0000	0.00
beta2	2nd db7	1.0000	0.00
beta2	2nd sym2	1.0000	0.00
beta2	3rd sym2	1.0000	0.00
beta2	4th sym2	1.0000	0.00

**Table 13 bioengineering-10-01074-t013:** The wavelet f1 score is the highest average f1 score based on the mother wavelet used for preprocessing with wavelet decomposition. The wavelet frequency band r is the mother wavelet (or no mother wavelet specified by None) that has most Spearman’s coefficients higher than the 95% percentile Spearman’s coefficient.

Wavelet f1 Score	Wavelet Frequency Band r
coif 4 (94.56)	db8 (Delta)
sym 3 (94.51)	None (Theta)
db4 (94.48)	db4 (Alpha)
Haar (94.45)	db2 (beta1)
coif 5 (94.41)	None (beta2)

## Data Availability

Not appliable.

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
