# Peer review of "REM Sleep Stage Identification with Raw Single-Channel EEG"

_bioengineering, 2023, doi:10.3390/bioengineering10091074_

Round 1
Reviewer 1 Report
In the conducted study, REM stages were classified using single-channel EEG signals. A deep neural network architecture was developed for this purpose. The following points are recommended to be addressed in the study:
1. The significance of the chosen EEG channel in determining the REM stage and why these specific EEG channels were selected should be emphasized.
2. A comparison table of the literature should be presented.
3. The obtained results do not clearly indicate which REM stages were classified, and the classification performance among different REM stages is not understood.
4. If only REM-NREM classification was conducted, this should be explicitly stated in the article and expressed in the abstract.
5. The "Table ??" references in the Results section should be corrected.
The study is generally understandable upon reading; however, minor grammar corrections and revisions are needed.
Reviewer 2 Report
The manuscript is well presented and contains a thorough description of the analyses. However, as if often in the case in engineering - a key aspect of the study is not described -identifying REM sleep compared to what? is the study finding REM sleep in data containing REM sleep and other sleep stages and/or REM sleep compared to wakefulness? Figure 1 helpfully shows some raw data - however it isn't clear what sleep stage this is from. It would be most helpful to see data from REM sleep and non-rem sleep and the decompositions, so the reader can get much more of an intuitive feel for what the analysis is doing.
tables often contain a wealth of information about what was used for the classifier. In the discussion can this be simplified and clearly explain to the reader which aspects of the signal and signal processing are most adept at the classification.
There are some v. minor formatting issues that need to be rectified - allusions to table ? - need to replace with the table number rather than a question mark.
Generally fine, just proof read and check sentences, tenses vary a bit - best to put everything in the past tense
Round 2
Reviewer 1 Report
The article is suitable for publication after the specified revision.